# Antimicrobial Efficacy against Antibiotic-Tolerant *Staphylococcus aureus* Depends on the Mechanism of Antibiotic Tolerance

**DOI:** 10.3390/antibiotics11121810

**Published:** 2022-12-13

**Authors:** Emily M. Meredith, Lauren T. Harven, Andrew D. Berti

**Affiliations:** 1Department of Pharmacy Practice, College of Pharmacy and Health Sciences, Wayne State University, Detroit, MI 48201, USA; 2Department of Biochemistry, Microbiology and Immunology, College of Medicine, Wayne State University, Detroit, MI 48201, USA

**Keywords:** MRSA, *Staphylococcus aureus*, tolerance, persistence

## Abstract

Bacteria can adapt to a changing environment by adopting alternate metabolic states favoring small molecule synthesis and resilience over growth. In *Staphylococcus aureus*, these states are induced by factors present during infection, including nutritional limitations, host responses and competition with other bacteria. Isogenic “tolerant” populations have variable responses to antibiotics and can remain viable. In this study, we compared the capability of antibiotics to reduce the viability of *S. aureus* made tolerant by different mechanisms. Tolerance was induced with mupirocin, HQNO, peroxynitrite or human serum. Tolerant cultures were exposed to ceftaroline, daptomycin, gentamicin, levofloxacin, oritavancin or vancomycin at physiological concentrations, and the viability was assessed by dilution plating. The minimum duration for 3-log viability reduction and 24 h viability reduction were calculated independently for each of three biological replicates. Each tolerance mechanism rendered at least one antibiotic ineffective, and each antibiotic was rendered ineffective by at least one mechanism of tolerance. Further studies to evaluate additional antibiotics, combination therapy and different tolerance inducers are warranted.

## 1. Introduction

*Staphylococcus aureus* remains the most common invasive human pathogen causing 120,000 cases of invasive infection in the United States annually and over 20,000 infection-attributable deaths [1]. Antibiotic resistance is common in staphylococci resulting in limited treatment options, higher mortality and increased healthcare costs [2,3]. Although antibiotic resistance has long been considered a risk for treatment failure, antibiotic tolerance has recently also been recognized to play a significant role in negative clinical outcomes [4]. 

Antimicrobial tolerance occurs when a subpopulation of bacteria adopts a distinct physiological state that facilitates survival in the presence of antibiotic concentrations that are bactericidal to the larger population. Although both populations are genetically identical, the presence of certain genetic mutations can alter the prevalence and rates at which antibiotic-tolerant subpopulations emerge [5,6].

In addition to genetic changes favoring larger “tolerant” subpopulations, there is increasing evidence that tolerant states can be induced by factors present in the environment, particularly those encountered during infection [6]. Rather than a stochastic phenomenon affecting only a small subpopulation, exposure to such environmental factors can provoke an entire population to exhibit the tolerant phenotype. 

These triggers alter a broad number of bacterial physiological pathways that converge into a tolerant phenotype [7]. It remains unclear whether staphylococci induced to become tolerant by one triggering factor exhibit the same response to antibiotic insult as bacteria induced to become tolerant by a different trigger.

Most studies to date examining tolerance in *S. aureus* have focused on a single tolerance-inducing condition, employing different bacterial strains, different endpoints and only a subset of antistaphylococcal antibiotics, often at concentrations that are not clinically relevant. Therefore, it is important to systematically examine these mechanisms to assess the effectiveness of contemporary antistaphylococcal therapies under tolerance-inducing conditions. Our goal was to determine if antistaphylococcal antibiotics remain capable of reducing the viability of tolerant *Staphylococcus aureus* bacteria and if such capability is linked to the mechanism of tolerance induction.

## 2. Results

All antistaphylococcal antibiotics reduced the viability of non-tolerant bacterial cultures, frequently achieving bactericidal activity within 48 h (Table 1). For each antibiotic, there existed at least one tolerance-inducing condition that prevents bactericidal activity over 48 h (Figure 1). This revealed that none of the antistaphylococcal antibiotics assessed are consistently effective in the simulated in vivo environment. Of the antibiotics tested, oritavancin remained the most consistently active; however, it became completely ineffective under humoral tolerance conditions (Figure 2). In fact, daptomycin was the only agent that retained activity against humoral tolerance conditions, albeit with a slower time to bactericidal activity.

## 3. Discussion

Antibiotic tolerance is associated with negative clinical outcomes in patients with *S. aureus* bacteremia [8]. Unfortunately, several different environmental triggers and distinct metabolic pathways converge in the phenotype of antimicrobial tolerance. In this study, we simulated antimicrobial tolerance by four well-defined inducing conditions and determined the impact on antistaphylococcal antibiotic activity.

Nutritional tolerance is a conserved and well-described microbial response to sudden nutritional deprivation [9,10]. An acute reduction in the amino acid availability results in an increase in uncharged transfer RNAs, which are recognized by the *rel* system [11,12]. An activated *rel* system produces the alarmone ppGpp and initiates a stringent response that is characterized in *S. aureus* by impaired ribosome assembly, slow growth and antimicrobial tolerance [13]. 

Mupirocin mimics this process via isoleucyl tRNA synthetase inhibition resulting in the sudden accumulation of uncharged isoleucyl tRNA and stringent response activation [14]. In the current study, nutritional tolerance conferred cross-tolerance to all antistaphylococcal antibiotics (Table 1, Figure 1). However, oritavancin does retain bactericidal activity against nutritionally-tolerant staphylococci, albeit with a statistically-significant delay in time to bactericidal activity. The clinical significance of such a delay in the context of an antibiotic with such a prolonged half-life is unclear.

Competitive tolerance is a common consequence of co-culture. Even within the same taxa, microbes often produce toxin-antitoxin systems or metabolic inhibitors to suppress the growth of “not-self” bacteria [15]. One such system involves the staphylococcal cytochrome bc1 inhibitor HQNO, which (when produced by *Pseudomonas aeruginosa*) impairs *S. aureus* growth by impairing electron transport [16]. This forces staphylococci to adopt fermentative growth and dissipate the proton-motive force. As aminoglycosides require an active proton motive force for bacterial internalization [17], it is no surprise that, in the current study, competitive tolerance completely blocked gentamicin and, to a degree, levofloxacin but did not impact antibiotics that act extracellularly, such as ceftaroline and oritavancin (Table 1).

Peroxynitrite is a potent oxidant produced by the activated macrophage oxidative burst and may be the primary reactive oxygen species at the site of infection [18]. Bacterial aconitase is exquisitely susceptible to redox damage and is specifically deactivated by peroxynitrite [19]. Aconitase inhibition results in the collapse of the bacterial TCA cycle and forced fermentative growth. However, in this form of fermentative growth, a proton-motive force can still be generated, albeit without the benefit of the TCA cycle generation of reducing equivalents. Therefore, although both competitive and oxidative tolerance result in fermentative growth, they do not induce the same pattern of tolerance to individual antibiotic agents, such as gentamicin (Figure 1C).

Humoral tolerance results from the activation of the staphylococcal GraSR cell envelope stress regulon [20,21]. This results in the downregulation of cell wall hydrolases and thickening of the bacterial cell wall [22,23]. The inducer of humoral tolerance, human cathelicidin (LL-37), is present at basal levels in the bloodstream but can also be produced by neutrophils in response to bacterial infection [24]. Even basal levels of cathelicidin are capable of inducing humoral tolerance sufficient to render most antistaphylococcal agents ineffective. 

Of all the forms of tolerance tested, humoral tolerance was the only one capable of rendering oritavancin ineffective (Figure 2E). Whether this oritavancin tolerance is a consequence of GraSR-mediated cell wall thickening or the modulation of a different member of the GraSR regulon is unclear. The only agent that retained some potency against humorally-tolerant *S. aureus* was daptomycin, albeit with a significantly prolonged time to bactericidal activity (Figure 2B).

This study has several limitations. Although we have tested and validated each of these forms of tolerance in other clinical isolates, we only performed the systematic analysis with one pan-susceptible *Staphylococcus aureus* strain. By necessity, this study was conducted in vitro in the absence of innate and adaptive immune responses. The simulations were only observed for a 48 h period and, thus, may fail to capture differences in activity beyond this timepoint. Finally, the antibiotics were added at static concentrations, which does not adequately recapitulate the pharmacokinetics and pharmacodynamics of clinical dosing regimens.

## 4. Materials and Methods

### 4.1. Strain Characterization, Cultivation Conditions and Antibiotic Selection

The prototypical methicillin-susceptible *Staphylococcus aureus* strain SH1000 was selected for analysis due to its susceptibility profile and its prior use in antibiotic tolerance studies [23,25]. Bacteria were cultivated in Mueller–Hinton 2 broth (BD Difco, Franklin Lakes, NJ, USA) unless otherwise noted. Antibiotics were supplemented at static concentrations corresponding to the estimated free serum concentration following standard dosing, including ceftaroline (Allergan, Madison, NJ, USA, 17 mg/L), daptomycin (Mylan, Canonsburg, PA, USA, 6 mg/L), gentamicin (Alfa Aesar, Haverhill, MA, USA 5 mg/L), levofloxacin (Teva Pharmaceuticals, Tel Aviv, Israel, 4 mg/L), oritavancin (Melinta, New Haven, CT, USA, 14 mg/L) and vancomycin (Mylan, New Haven, CT, USA, 35 mg/L). 

Mupirocin was obtained from Panreac AppliChem, Chicago, IL, USA. Active ceftaroline was generated from ceftaroline fosamil by enzymatic amino dephosphorylation [26], and we validated its activity by bioassay prior to use. All media containing oritavancin were supplemented with Tween20 (0.002%), and all media containing daptomycin were supplemented to 50 mg/L Ca^2+^ per CLSI recommendations [27]. Human male blood type AB serum was obtained from Sigma Aldrich, St. Louis, MO, USA. Antibiotics were added to human serum at static concentrations corresponding to the estimated total serum concentration following standard dosing as follows: ceftaroline, 21 mg/L; daptomycin, 80 mg/L; gentamicin, 5 mg/L; levofloxacin, 6 mg/L; oritavancin, 16 mg/L plus 0.002% Tween20; and vancomycin, 150 mg/L.

### 4.2. Nutritional Tolerance

Acute nutritional limitation was simulated by the method of Reiss et al. [28]. Briefly, overnight bacterial cultures were adjusted to approximately 1 × 10^8^ colony forming units (cfu) per mL with fresh Mueller–Hinton broth. Cultures were then supplemented with mupirocin at a concentration of 3.2 mg/L and returned to the incubator with shaking for 1 h (37 °C, 180 rpm), after which antibiotics were added at the above concentrations.

### 4.3. Competitive Tolerance

Competition with *Pseudomonas aeruginosa* was simulated by the method of Orazi et al. [16]. Briefly, overnight bacterial cultures were adjusted to approximately 1 × 10^8^ cfu/mL with fresh Mueller–Hinton broth. Cultures were then supplemented with 2-heptyl-4-hydroyquinoline-N-oxide (HQNO) to a final concentration of 3 mg/L and returned to the incubator with shaking for 1 h (37 °C and 180 rpm), after which antibiotics were added at the above concentrations.

### 4.4. Oxidative Tolerance

Exposure to tolerance-inducing reactive oxygen species was simulated by the method of Beam et al. [19]. Briefly, overnight bacterial cultures were adjusted to approximately 1 × 10^8^ cfu/mL with fresh Mueller–Hinton broth. Cultures were then supplemented with peroxynitrite to a final concentration of 2 mM and returned to the incubator with shaking for 1 h (37 °C and 180 rpm), after which antibiotics were added at the above concentrations.

### 4.5. Humoral Tolerance

Cathelicidin-mediated tolerance was simulated by the method of Ledger et al. [23]. Briefly, approximately 1 × 10^8^ cfu/mL bacteria were inoculated into human serum and incubated with shaking overnight (~16 h, 37 °C, 180 rpm). Antibiotics were then added at the above concentrations. Parallel experiments using freshly-inoculated bacteria added to separate tubes concomitant with antibiotics were included as non-serum-adapted controls.

### 4.6. Data Analysis

Samples were removed for colony enumeration immediately prior to the addition of antibiotics and at set intervals after antibiotic challenge. Samples were enumerated by dilution plating on brain heart infusion agar (BD Difco, Franklin Lakes, NJ, USA). Antibiotics were considered “bactericidal” if the viable cfu/mL decreased from the baseline by more than 3 log_10_ units over a 48-h period. The minimum duration to bactericidal activity (MDK_99.9_) [4] was determined individually per replicate via linear extrapolation between the timepoints immediately preceding and following a 3 log_10_ unit reduction from baseline. All analyses were performed in triplicate.

Descriptive data were expressed as the mean and standard deviation. Differences in MDK_99.9_ between tolerance-exposure conditions were determined by ANOVA with post hoc Student’s *t*-test, including Holm–Bonferroni adjustment. A *p*-value ≤ 0.05 was considered to be significant.

## 5. Conclusions

The mechanism by which antimicrobial tolerance is induced impacts both the time to bactericidal effect and the extent of killing. No antibiotic was consistently bactericidal against all forms of tolerance; conversely, no form of tolerance was able to render all antistaphylococcal antibiotics ineffective. Further studies are needed to evaluate the potency of additional antistaphylococcal antibiotics, antibiotic activity in the context of different environmental inducers of tolerance and strategies to counteract the establishment of antibiotic tolerance.

## Figures and Tables

**Figure 1 antibiotics-11-01810-f001:**
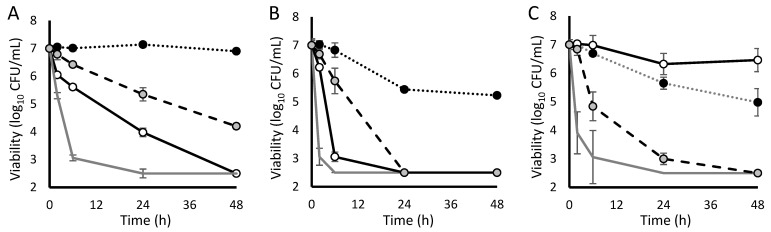
Activity of study antibiotics against induced-tolerant staphylococci determined in Mueller–Hinton broth. (**A**) ceftaroline; (**B**) daptomycin; (**C**) gentamicin; (**D**) levofloxacin; (**E**) oritavancin; and (**F**) vancomycin. Dotted line with black points, nutritional tolerance; dashed line with gray points, oxidative tolerance; solid line with white points, competitive tolerance; and gray line without points, no tolerance induction.

**Figure 2 antibiotics-11-01810-f002:**
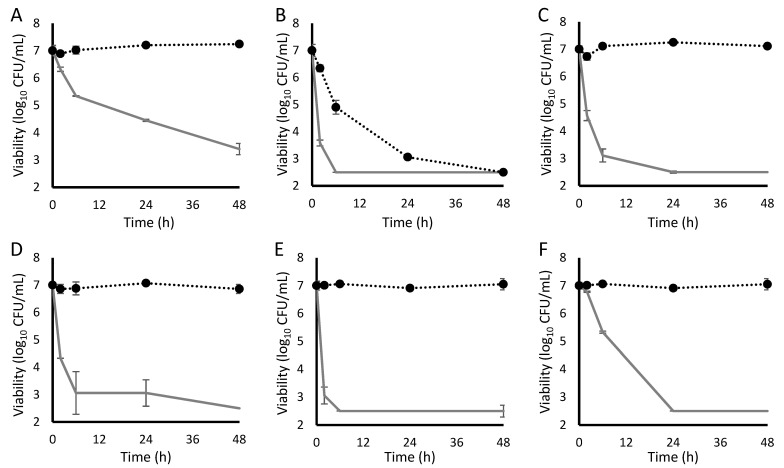
Activity of study antibiotics against induced-tolerant staphylococci determined in human serum. (**A**) ceftaroline; (**B**) daptomycin; (**C**) gentamicin; (**D**) levofloxacin; (**E**) oritavancin; and (**F**) vancomycin. Dotted line with black points, bacteria conditioned overnight in serum prior to antibiotic exposure; and gray line without points, bacteria inoculated into serum immediately prior to antibiotic exposure.

**Table 1 antibiotics-11-01810-t001:** Time needed to induce a 3-log reduction in viability (hours).

Antibiotic	No Induction	Nutritional Tolerance	Competitive Tolerance	Oxidative Tolerance	Humoral Tolerance
Ceftaroline	11.1 ± 0.19	†	11.1 ± 1.05	47.6 ± 0.34 *	†
Daptomycin	1.5 ± 0.25	†	4.4 ± 0.09 *	17.7 ± 0.95 *	12.9 ± 0.72 *
Gentamicin	1.3 ± 0.17	23.4 ± 0.02 *	†	21.9 ± 1.95 *	†
Levofloxacin	1.7 ± 0.13	†	4.9 ± 0.56 *	31.8 ±3.00 *	†
Oritavancin	1.2 ± 0.08	5.3 ± 0.41 *	1.2 ± 0.02	1.4 ± 0.12	†
Vancomycin	25.8 ± 6.12	†	†	45.2 ± 1.39 *	†

† 3-log reduction in viability was not achieved within 48 h. * *p*-value < 0.01 vs. uninduced control.

## Data Availability

The data presented in this study are available on request from the corresponding author.

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
