# Peer review of "Antimicrobial Efficacy against Antibiotic-Tolerant Staphylococcus aureus Depends on the Mechanism of Antibiotic Tolerance"

_antibiotics, 2022, doi:10.3390/antibiotics11121810_

Round 1

Reviewer 1 Report

Harven et al present a report on the ability of antibiotics primarily used in the treatment of Staphylococcus aureus infections to reduce the viability of tolerant S. aureus. The study is of a high clinical relevance, given that S. aureus is a well-recognized global health threat and has a high propensity for developing resistance to antimicrobials. The manuscript is well written and would be of interest to clinicians and researchers working in anti-staphylococcal therapy. The authors, however, need to address the following concerns to improve the quality of the manuscript:

1. The Introduction section could use a bit more information on the clinical significance of S. aureus. Also, the novelty of the study needs to be further accentuated.

2. The authors may want to reposition the table and figures, such that they appear within the Results section, as that is where the “callouts” have been made. This would make it easier for readers to follow the findings the authors are communicating.

3. Abbreviations, such as “ORI”, “DAP”, “VAN”, etc., need to be defined the first time they are used. These issues are typically in the Results and the Discussion sections.

4. The authors may want to delete the “As expected” that begins the Results presentation, as its presence gives the impression of a discussion being done.

5. The presentation of the Discussion section looks more like a general review of the literature, rather than a contextualization of the findings within what is known in the study area. The findings being discussed appear obscured and need to be better articulated.

6. At parts of the manuscript where the authors have the citations as parts of the sentences, such as in Lines 150, 156, 164, and 169, the citations need to terminate in the numbers alongside which the cited works appear in the bibliography. For example, Reiss et al. should be written as “Reiss et al. [26]”, rather than placing the “[26]” at the end of the paragraph.

7. In the Materials and Methods section, there are issues with the numbering of the subsections, specifically, the last two subsections – “4.2 Data Analysis” and “4.2 Statistical Analysis”. Besides, it may be better to combine these two subsections.

8. In the Statistical Analysis subsection, the authors need to specify what they sought to detect with the univariate analyses. Is it to detect differences in viability, tolerance indicators, etc.?

9. Could the authors kindly provide details regarding the source of the culture media used?

10. There are some issues with grammar, and the authors may want to re-check the manuscript for such errors. A few of these are outlined below:

a. Line 12: Kindly make the “environment” plural.

b. Line 16: Kindly write the “compare” in the past tense, as the manuscript is a report. This comment applies to other sections that need to be in reported speech, but have been presented in the present tense, such as in Line 48, where “is” needs to be replaced with “was”.

c. Line 25: Please italicize the “Staphylococcus aureus” that appears among the keywords.

d. Line 33: Please replace the “for” in “bactericidal for the larger population” with “to”.

e. Line 39: Regarding the “effecting”, do the authors mean “affecting”?

f.  Lines 42 to 44: The clarity of the sentence needs to be further enhanced. For example, the authors could choose to move the “remains unclear” that ends the sentence to the beginning of the sentence and precede it with “It”, i.e., “It remains unclear whether staphylococci…”.

g. Line 48: The authors may want to rewrite “to reduce” as “in reducing”.

Author Response

Please see attachment and thank you for an excellent critique. We hope that the changes we have made address your concerns.

Reviewer 2 Report

Dear authors,

Although the subject is very interesting, I suggest the manuscript need extensive work.

The introduction section could provide more background and there are more articles in the literature to cite om the subject.

I strongly believe the methods; the research design and the result sections must be rewritten.

The chosen subject is very interesting and important, and I believe you should reorganise your work.

Author Response

We thank the Referee for his or her expert review of our manuscript. We have increased the number of citations in the introduction section and revised some of the content. We have updated the methods, design and result sections consistent with the specific requests provided by the other Referee. In the absence of specifics, we hope this addresses the reorganization requested by this Referee.

Round 2

Reviewer 2 Report

Thank you